# Chemosensory and Behavioural Responses of *Ixodes scapularis* to Natural Products: Role of Chemosensory Organs in Volatile Detection

**DOI:** 10.3390/insects11080502

**Published:** 2020-08-04

**Authors:** Nicoletta Faraone, Michael Light, Catherine Scott, Samantha MacPherson, N. Kirk Hillier

**Affiliations:** 1Department of Chemistry, Acadia University, Wolfville, NS B4P 2R6, Canada; 2Department of Forestry, University of Toronto, Toronto, ON M5S 1A1, Canada; mikelight@acadiau.ca; 3Department of Biology, Acadia University, Wolfville, NS B4P 2R6, Canada; catherine.scott@acadiau.ca (C.S.); 133207m@acadiau.ca (S.M.); kirk.hillier@acadiau.ca (N.K.H.)

**Keywords:** tick electrophysiology, *Ixodes scapularis*, chemosensory organs, attractant, deterrent, terpenoids, scutum, inhibitory, mixture interactions, VOCs

## Abstract

**Simple Summary:**

Ticks are responsible of transmitting serious disease agents of importance to human and veterinary health. Despite the importance of repellents, deterrents and acaricides in tick management, little is understood about the types of chemicals recognized and the mechanism behind chemoreception. Being almost totally blind, ticks rely on chemosensation to identify and locate hosts for a successful blood meal and to detect chemical signals in the environment. We explored the neurophysiology of tick chemosensory system in the context of behaviourally-relevant volatile stimuli, including essential oil components, to evaluate how the combination of attractants and plant volatile compounds is detected and processed. The observed inhibition (or deterrent effect) in tick electrophysiological response and behavioural activity, after the tick has been exposed to a binary mixture of attractant and volatile compound, represents an important advancement in understanding how tick olfaction works and what may be the mechanism behind detecting unpleasant odor stimuli and consequently been deterred. These information will provide more insights in developing new natural product-based deterrents for self-protection.

**Abstract:**

Blacklegged ticks, *Ixodes scapularis*, represent a significant public health concern due to their vectoring of tick-borne disease. Despite their medical importance, there is still limited knowledge of the chemosensory system of this species, and thus a poor understanding of host-seeking behaviour and chemical ecology. We investigated the electrophysiological sensitivity of adult female blacklegged ticks to attractants and plant-derived compounds via an electrode inserted into the scutum. The response of female ticks to binary mixtures with a constant concentration of a selected attractant (butyric acid) and increasing concentration of volatile organic compounds (VOCs) (geraniol, phenethyl alcohol, β-citronellol, and citral) was recorded. A strict relationship between increasing volatile concentration and a decreasing response was observed for each VOC. Y-tube bioassays confirmed that tick attraction towards butyric acid decreased with the presence of a VOC, which exerted a deterrent effect. To determine the specific role of sensory appendages involved in the detection of attractant chemical stimuli, we tested tick electrophysiological response after removing appendages that house chemosensory sensilla (foretarsi, pedipalps, or both). The chemosensory response was related to the molecular structure of attractant odorant, and the lack of pedipalps significantly reduced olfactory responses, suggesting they play an important role in detecting attractants. This study provides new insight into the neurophysiological mechanisms underlying tick olfaction and the potential for interactions between attractant and deterrent chemical detection.

## 1. Introduction

Ixodid ticks are bloodsucking vectors of serious pathogens of medical and veterinary importance to humans and other vertebrates. They find their hosts by cues derived from semiochemicals that act as attractants or repellents [1]. The tick chemosensory system represents a new frontier for scientific exploration if compared with that of insects [2,3]. The lack of antennae, the presence of very few olfactory sensilla, the development of a distinctive organ (Haller’s) used for volatile detection, and the multimodal role of sensilla located on mouthparts make ticks a fascinating system for understanding how chelicerates (and particularly Ixodidae) detect chemicals [4,5,6]. Previous studies have focused on investigating tick responses to attractant stimuli, such as chemicals emitted by suitable hosts (CO_2_, typical human odors, or compounds produced by vertebrates) [7,8,9,10,11,12,13]. Moreover, there are several examples of behavioural bioassays used to validate the repellent activity of compounds from synthetic and natural origin, allowing isolation of potential new active ingredients for the development of more effective tick repellent products [13,14]. However, most repellency assays for ticks do not discriminate between repellency and deterrence due to olfaction versus that from contact chemoreception [15], and evidence suggests that olfaction is involved at least in part in tick responses to repellents [13].

The mechanisms that ticks use to detect odor stimuli and the behavioural function of such stimuli as repellents, deterrents or attractants are not well understood [3]. There are few examples attempting the use of traditional electrophysiological methods that measure the chemosensory response after the delivery of a specific stimulus [16]. Single sensillum recording (SSR) has been used to target one of the few sensilla housed in the Haller’s organ capsule [17,18,19]. In another study, the use of a glass capillary electrode filled with a saline solution inserted on the exposed synganglion was used to correlate tick questing behaviour in relation to odors [20]. Few studies have investigated the role of sensilla housed in the mouthparts in chemosensory detection. Gustatory sensilla (“palpal receptors”) [21] are located on the pedipalps of ticks and there is evidence for detection from electrophysiological and behavioural studies [22,23,24]. The palpal organs are apparently not critical for detecting aggregation pheromone [25]. However, sensilla located on the pedipalps are responsible for short-range chemical detection of repellents [13], along with other non-volatile compounds, thus they seem to function as multimodal sensory organs that complement the long-range sensory function of the Haller’s organ.

To our knowledge, a limited number of studies have been published investigating tick repellents via electrophysiology [20,26], and none have discretely evaluated dose-response relationships. More studies are required on chemosensory system-repellent compound interaction at a neurological level in order to understand the mechanism behind repellency and chemosensory detection. This information is important for the development of personal tick repellent formulations effective in reducing exposure to tick bites and pathogen transmission [27].

In this study, we examined the electrophysiological response of the chemosensory system of adult female blacklegged ticks (*Ixodes scapularis*, Say 1821) in response to plant-derived compounds with known repellent activity [13], and to known volatile attractants [19] using a novel electrophysiological approach we called “electroscutumography” (ESG). We tested the response to attractants (aldehydes and carboxylic acids with increasing hydrocarbon chain length) and volatile organic compounds (VOCs) previously reported as repellents, such as geraniol, phenethyl alcohol (PEA), citral, and β-citronellol [13]. In order to reproduce a typical “field condition” where ticks are exposed to vertebrate attractants together with VOCs (e.g., human body odor mixed with repellent products), we recorded the electrophysiological response of binary mixtures with a constant concentration of a selected attractant, butyric acid (BTA), and increasing concentrations of VOCs. We then performed behavioural bioassays using a Y-tube olfactometer, to assess how a tick’s attraction towards BTA might be impacted by the presence of a putative repellent compound. In addition, to determine the specific role of chemosensory organs involved in chemical detection, we tested tick electrophysiological response after selectively removing appendages that house chemosensory sensilla (i.e., foretarsi, pedipalps, or both). This study improves our understanding of tick chemoreception and potential interactions between VOCs in affecting tick host-seeking behaviour.

## 2. Materials and Methods

### 2.1. Ticks

Naïve, unfed, host-seeking (actively questing) adult female *I. scapularis* ticks were used in all behavioural repellency trials and electrophysiological recordings. Each tick was only used in one experiment. Uninfected ticks were purchased from the Tick Rearing Facility Laboratory at Oklahoma State University (Oklahoma, USA). Ticks were stored on-site in plastic containers lined with moistened Kimwipe^®^ in the fridge, at 4 °C in dark conditions. Ticks were considered actively questing/host-seeking if they raised their forelegs or began crawling when exhaled upon by an investigator.

### 2.2. Chemicals

Geraniol (>98%), phenylethyl alcohol (≥99%, FCC, FG), citral (>95%), β-citronellol (≥95%), propionic acid (≥99.5%, FCC, FG), isovaleric acid (≥99%), hexanal (98%), octanal (≥99%), hexanoic acid (≥99%) were purchased from Sigma-Aldrich (Saint Louis, MO, USA). Butyric acid (≥98%), lactic acid (≥98%), isobutyric acid (≥98%), butanal (≥99%), pentanal (≥99%) were purchased from Bedoukian (Danbury, CT, USA). Ammonia (≥99%) was purchased from Fisher Scientific (Waltham, MA, USA), and heptanal (92%) from SAFC (Steinheim, Germany). VOCs [13,28] and attractant [12,19] compounds were selected based on previous studies reporting their behavioural effect.

In electrophysiological tests, the concentration of VOC stimuli was selected based on preliminary dose-response assays and it ranged between 100–1000 μg/μL in hexane (Sigma-Aldrich, hexane CHROMASOLV, >98.5% purity). The concentration of the attractant (BTA) was constant at 100 μg/μL (Table 1). This range of concentrations was selected based on preliminary dose-response experiments performed in order to select the minimum amount of stimulus required to elicit an electrophysiological response significantly different from the control stimulus (hexane).

In the Y-tube behavioural bioassay, PEA was evaluated for repellent action at 1 μg/μL, and BTA was evaluated as attractant at 1 μg/μL, as selected based upon preliminary dose-response experiments.

### 2.3. Electroscutumography (ESG) Recording

Prior to the beginning of the ESG recordings, unfed adult tick females were set outside the fridge for a minimum of 20 min allowing the tick to acclimate to room temperature. An individual tick was placed on a glass slide covered with a thin layer of dental wax, head on facing the stimulus airflow (Figure 1). The first pair of legs (bearing the Haller’s organs) were mounted on a small piece of shattered glass as a platform. The whole tick body was held still by a minuten pin crossing the tick body horizontally (ENTO SPHINX, Černá za Bory, Czech Republic).

Using a sharp syringe needle (PrecisionGlide^®^ 21G1), a small incision was made on the scutum in the approximate location of the synganglion. Tungsten electrodes, prepared using a 0.13 × 76 mm tungsten rod electrolytically sharpened to approximately 1 µm in saturated potassium nitrite solution, were used for both recording and ground electrodes. Electrode gel (SIGNAGEL, Parker Laboratories Inc., Fairfield, NJ, USA) was placed on the sharpened electrodes before establishing a contact to improve connection. The ground electrode was inserted in the tick’s lower body. The recording electrode was inserted directly into the incision previously made on the scutum and in contact with the soft tissue bearing the synganglion. The insertion of the electrodes was done under a Nikon SMZ645 microscope (Microscope Central, Feasterville, PA, USA), and the responsiveness of the electrophysiological preparation was confirmed when a response was observed after human breath. This mounting set-up was adapted from Romanshchenko et al. [20]. A constant humidified airflow (0.5 L/min) was delivered to the tick preparation through a 30 cm long glass air delivery tube with a single 2 mm diameter hole 10 cm from the outlet. A Syntech Intelligent Data Acquisition Controller-2 (IDAC-2) system was used to collect and amplify changes in electrical potential (Low Cut-off: 0.05 Hz, Offset: 0, Ext amp: 10; Ockenfels SYNTECH GmbH—Buchenbach, Germany).

Stimulus cartridges were made by applying 10 μL of the test solution to a 1 x 5 cm^2^ piece of filter paper (Fisherbrand^®^, Pittsburgh, PA, USA) and left to dry out under a fume hood for 5–10 min. For stimuli having both attractant and repellent, both compounds were individually loaded on the same cartridge. Loaded cartridges were then individually inserted in 3-mL-capacity glass Pasteur pipettes secured at both ends with 1000 μL pipette tips (Fisherbrand^®^, Pittsburgh, PA, USA). New stimulus cartridges were prepared after every two uses. A test stimulus was delivered during 0.5 s in an air puff while the tip of Pasteur pipette was inserted through the hole in the air delivery tube. Stimuli were presented at 1 min intervals to allow the olfactory system to return to baseline rates between puffs.

In attractant-VOC blends each constituent was presented in increasing concentration steps, beginning with 100 μg of BTA, and increasing concentration of VOC (with 100 μg of BTA constant), up to 1000 μg. Stimulus order (geraniol, PEA, β-citronellol, citral) was provided in a random fashion, and each tick (*N* = 21) was presented with a control stimulus (hexane) at the beginning and end of the recording and between presentations of each VOC (5 hexane puffs total).

For the attractant series (constant concentration of 100 μg/μL), the stimulus order was randomised. For each series of stimuli, a control stimulus (hexane) was delivered at the beginning, in the middle, and at the end of the sequence. The full series of attractants was presented to intact ticks, ticks with foretarsi removed, ticks with pedipalps removed, and ticks with both pairs of appendages removed (*N* = 21 ticks in each treatment group). The surgical removal of pedipalps and tarsi was performed as previously described [13].

### 2.4. Y-Tube Behavioural Bioassay

We used a vertically oriented glass Y-tube olfactometer (diameter: 4 cm; entry arm: 20 cm; choice arms: 10 cm) to test behavioural responses of unfed adult female ticks to volatile chemical stimuli. Humidified air was pumped through the arms of the Y at a pressure of 17.5 psi using a 2-channel clean air delivery system (Sigma Scientific Model CADS-2P). A Y-shaped wire apparatus (diameter: 1 mm) was inserted into the glass Y-tube as a substrate on which ticks could climb. This included a 1-cm “questing zone” at the decision point on which ticks could rest and test the air from each arm before making a choice. Test stimuli were applied to pieces of filter paper (one-quarter of a Whatman No. 1 125 mm filter paper circle folded in half and stapled into a cone) which sat on the ends of each choice arm of the climbing wire at the distal ends of the Y-tube. After filter papers were loaded with test stimuli, they were left for ~5 min to allow the hexane to evaporate. Questing ticks were allowed to climb onto a bamboo skewer, then the end of the skewer was placed in contact with the climbing wire just above its attachment point to the Y-tube. The tick was then allowed to climb up the skewer and onto the climbing wire. If a tick did not climb up to the decision point of the Y within 5 min, it was considered a non-responder. A choice was recorded once a tick left the decision point and climbed at least 8 cm up the choice arm toward the stimulus. The locations of the treatment and control stimuli (left of the right arm of the Y-tube) were alternated between replicates. The wire climbing rod was washed with soap and water and then ethanol between replicates to eliminate any contact chemical cues left by climbing ticks.

Experiment 1 (attractant vs. control)—Ticks (N = 16) had a choice between an attractant stimulus (10 μL of 1 μg/μL BTA in hexane) and a control stimulus (10 μL of hexane).

Experiment 2 (attractant + VOC vs. control)—Ticks (N = 16) had a choice between an attractant stimulus together with a putative repellent stimulus in a 1:1 ratio (10 μL of 1 μg/μL BTA in hexane plus 10 μL of 1 μg/μL PEA in hexane) and a control stimulus (20 μL of hexane).

### 2.5. Statistical Analysis

General statistical methods—We ran all statistical analyses in Rstudio using R 4.0 [29]. We log-transformed variables that appeared log-normal and confirmed that the transformation resulted in assumptions of normality being satisfied based on visual inspection of residual plots for each model. After running models on transformed variables, we back-transformed calculated marginal means and 95% confidence intervals and report these on the original scale of measurement in the results. For all analyses, we used an α of 0.05, and we used Tukey’s method for calculating adjusted *p*-values for post-hoc multiple comparisons.

ESGs with attractants—We assayed which volatile compounds elicited ESG responses stronger than responses to hexane control puffs (averaged over three hexane puffs for each tick) using only the data from intact ticks. We log-transformed the ESG response amplitude and used this as the response variable in a linear model with volatiles as a fixed effect and tick number as a random effect, to account for each tick being exposed to each stimulus. We then used Tukey’s method to make post-hoc comparisons of responses to each of the tested VOCs.

For all compounds for which responses were significantly stronger than to hexane, we then asked whether surgical removal of the foretarsi, palps, or both affected ESG responses. We calculated a relative response by dividing each response to a volatile compound (in mV) by each tick’s average response to hexane (in mV) resulting in a unitless value indicating how many folds greater the intensity of each response was than responses to hexane. We log-transformed this variable and used it as the response variable in a linear model with VOC surgical treatment, and their interaction as fixed effects. Since the interaction was not significant, we re-ran the model without it, and then used Tukey’s method to make post-hoc comparisons of the surgical treatments, averaged across all compounds. Following this, we ran one-way ANOVAs (and Tukey’s post hoc tests when ANOVAs resulted in significant effects) for each compound, to investigate the effects of surgical treatment on responses to individual compounds.

ESGs with VOCs plus butyric acid—We ran separate analyses for each volatile compound (PEA, geraniol, β-citronellol, and citral) to compare tick ESG responses to different concentrations of repellent and BTA (see Table 1). We first calculated relative response values (by dividing responses of each tick by its mean response to five hexane puffs, at the beginning and end of each run and between exposure to each compound). We then ran four linear models, each with a log-transformed relative response as the response variable, concentration of the compound (0, 100, 500, or 1000 μg) as a categorical fixed effect, and tick number as a random effect. We then used Tukey’s method to make post-hoc comparisons of responses to each concentration for each volatile.

Y-tube data—For ticks given a choice between two stimuli in Y-tube assays, we used binomial tests to ask whether they chose to enter one of the stimulus arms more often than expected by random chance (null expectation = 50% of ticks choosing treatment).

## 3. Results

### 3.1. Electroscutumography (ESG)

Responses to attractants—Unfed adult female *I. scapularis* ticks (*N* = 21) were exposed to a series of stimuli and responses were consistent throughout repeated presentations of the same stimulus to a given tick preparation. Eight of the attractant stimuli elicited a response significantly different to the control (Full model: F_12,168_ = 95.3, *p* < 0.001; Table 2) showing a potential correlation between chain length and functional groups on the VOC (Table 2). Interestingly, VOCs having between 3 and 6 carbons with functional groups as carboxylic acid or aldehyde elicited a response different from the control (hexane). The electrophysiological response increased with the decrease of the number of carbons, going from hexanal (6 carbons, 5.0 mv) to butanal (4 carbons, 8.4 mv), and from hexanoic acid (6 carbons, 6.6 mv) to propionic acid (3 carbons, 26.4 mv). Although with 3 carbons, lactic acid was not significantly different from the control, along with pentanal, an aldehyde with 5 carbons.

Responses to attractants with removed chemosensory organs—ESGs carried out after removing appendages that house chemosensory organs revealed that sensilla located on the pedipalps, together with those on the Haller’s organ, are responsible for detecting volatile attractant compounds (F_3,469_ = 11.26, *p* < 0.001). The surgical amputation of both pedipalps and foretarsi significantly decreased the detection of volatiles, as did the removal of the pedipalps alone. Ticks missing only the foretarsi were intermediate, suggesting that the palps are particularly important for detecting these attractants (Figure 2a).

When each compound is considered alone, similar patterns emerge for hexanoic acid, butanal, and isovaleric acid, whose detection was impacted by the surgical removal of both chemosensory organs (Figure 2e–g). The non-significant effects of surgery for the other odorants are likely influenced by high variation in response amplitude among ticks resulting in relatively low statistical power.

Mixtures of butyric acid with increasing concentrations of VOCs—Tick responses to BTA (a known attractant) were significantly impacted by the presence of VOCs (PEA: F_3,60_ = 59.7, *p* < 0.001; geraniol: F_3,60_ = 78.8, *p* < 0.001; β-citronellol: F_3,60_ = 173.5, *p* < 0.001; citral: F_3,60_ = 254.3, *p* < 0.001), showing a decrease in the neurological activity in correlation to increase of compound (Figure 3). All four selected terpenoids reduced tick responses relative to BTA alone. The greatest effect was seen for PEA: the highest dose reduced the relative response from 14.1-fold greater than the response to hexane to 4.1-fold greater (Figure 3a). The weakest effect was for citral: the highest dose reduced the relative response from 8.8- to 3.4-fold greater than to hexane (Figure 3d).

### 3.2. Behavioural Bioassays

Experiment 1 (attractant vs. control)—Ticks were significantly more likely to choose the olfactometer arm containing BTA than the hexane control (80% of responding ticks chose the treatment arm; binomial test; *p* = 0.035), indicating that they were attracted to BTA. Only one of 16 ticks did not make a choice (Figure 4b).

Experiment 2 (attractant + VOC vs. control)—When the attractant BTA was mixed with PEA in a 1:1 ratio, ticks were equally likely to choose the treatment arm and the hexane control arm (50% of responding ticks chose the treatment arm; binomial test; *p* = 1.0). In this test, four of 16 ticks did not make a choice (Figure 4c).

## 4. Discussion

Through a novel electrophysiological approach, we were able to measure the neurological response of blacklegged ticks to chemical stimuli [20]. The synganglion used to study tick response represents the central nervous system of ticks where all the chemosensory inputs are processed [5,30,31]. Tick sensory organs are different from the standard olfactory system present in insects [32]. Sensilla are reported to reside mainly in the Haller’s organ, located on the first segment of both foretarsi [5,19], while pedipalps together with chelicerae and the tip of the foretarsi were reported to house only taste receptors [22,24]. The surgical removal of pedipalps confirmed the versatile role of this organ, whose ability to detect volatiles was already reported in behavioural bioassays [13]. The absence of both Haller’s organ and pedipalps greatly reduced the electrophysiological response compared to intact ticks, suggesting that pedipalps are particularly important for detecting specific volatiles. The tick palpal receptors are tip-pore sensilla [21], similar to those present in spiders [33], which apparently function in olfaction in addition to gustation [34]. The observed response in ticks suggests that the tip-pore receptors might similarly be capable of both olfaction and gustation. This would be in contrast to insects, such as mosquitoes, in which the maxillary palp neurons house distinct gustatory and odorant receptors [35].

Tick electrophysiological responses revealed a stronger response as the length of the hydrocarbon chain decreases, and when a branched section or specific functional groups (i.e., aldehyde or carboxylic acid) are present, reflecting the differences in vapor pressure of presented compounds. The correlation between molecular structure and behaviour has been previously studied [36], although the relationship between the characteristic of a molecule and its biological activity is not yet clearly understood. As was previously observed for tick repellents and attractants, the chemical and physical features of odorant strongly mediate the ecological interaction of ticks, the result of which is crucial in regulating their behaviour [37].

In preliminary electrophysiological experiments carried out on VOC stimuli, no tick response was recorded as for the case of attractant compound (Table 2), and no dose-response correlation was observed by testing a different range of concentrations (0.1–1000 μg/μL) (data not shown). These observations led us to hypothesize that putative repellents might induce an inhibition in the tick olfactory system, consistently reducing the ability to detect chemical volatiles and act as deterrents [38]. The mode of action of repellents of blood-feeding arthropods is still somewhat controversial and several different hypotheses have been presented. For instance, a proposed mode of action of DEET, the “gold standard” of repellents, has been described as an inhibitor, interfering with the recognition of attractant odors, rather than a true repellent [39], or as a modulator of the general olfactory receptor activity and capable of disrupting the insect odor code [40,41]. In light of these findings, the behavioural response induced by some essential oil components to *I. scapularis* recorded in our previous study [13] may be better defined as a deterrent. Those volatile compounds (PEA, geraniol, citral, and β-citronellol) induced an effective avoidance behaviour by the recognition of these molecules by some specific tick olfactory receptors. However, in preliminary electrophysiological studies, they did not elicit any response, as they were apparently not detected by the tick chemosensory organs, suggesting a potential inhibitory action. In vitro assays with *I. ricinus* (L.), permethrin reported a similar response being active as a true deterrent only when paired with arrestment stimuli [37]. Differences between stimuli were significant (Figure 3), even though confidence intervals were wide probably because of the great variation in amplitude of response among tested ticks. The pattern in decreasing neurological response in relation to the increase of deterrent concentration is very consistent for each tick and it might be described as inhibitory action exerted by the compound on the tick chemosensory system. In a previous study carried out on heliothine moths, a molecular competition on mixture interactions and the effect of non-cognate odorants on the reception of ligands by pheromone sensilla was reported [42], suggesting a correlation with this mode of action exerted by a binary mixture attractant-repellent.

This observation was supported by behavioural bioassays conducted on ticks where an attractant odor or a binary mixture of an attractant and a VOC was offered in a Y-tube set-up. The presence of a volatile compound (PEA) disrupts the significant attraction exerted by BTA towards ticks that equally chose between the control arm and the treatment arm (Figure 4). Bioassays conducted in presence of tick attractants associated with a live host are usually more effective in forecasting the efficacy of the product in disrupting chemosensory perception [26,37,43], considering the fact that Y-tube olfactometers are probably not suitable for testing only repellents or deterrents [44]. In previous behavioural tests conducted in our laboratory, ticks needed to perceive the presence of the observer in order to activate locomotory behaviour. More replications with different compounds need to be performed, and different ranges of concentrations need to be tested to better clarify this inhibitory dose-response relationship on tick behavioural responses.

In this study, we explored the olfactory neurophysiology of *I. scapularis* in the context of behaviourally-relevant volatile stimuli using a novel ESG technique. Moreover, we evaluated the tick electrophysiological response towards attractants and how the combination of attractants and deterrents may impact the tick chemosensory system. This study provides insight regarding the physiological and behavioural responses of ticks to volatile stimuli such as plant essential oil components and body odor attractants. The novel observation of inhibition in tick electrophysiological activity by deterrents represents an important advancement in understanding how tick’s olfaction works and the mechanism underlying their responses to deterrent stimuli.

## 5. Conclusions

Novel dose-response inhibition in tick electrophysiological activity was observed, in concert with tick exposure to deterrent and attractant compounds. This was also confirmed through Y-tube behavioural bioassays showing a significant decrease in attraction towards the stimulus when a deterrent was present. The removal of the appendages that bear chemosensory sensilla validated the important role of pedipalps in detecting volatile compounds. The electrophysiological response might be impacted by the chemical structure and functional groups present in the compound, influencing the detection and longevity of attractant and deterrent action. More electrophysiological and behavioural investigations need to be performed in order to confirm and corroborate these findings.

## Figures and Tables

**Figure 1 insects-11-00502-f001:**
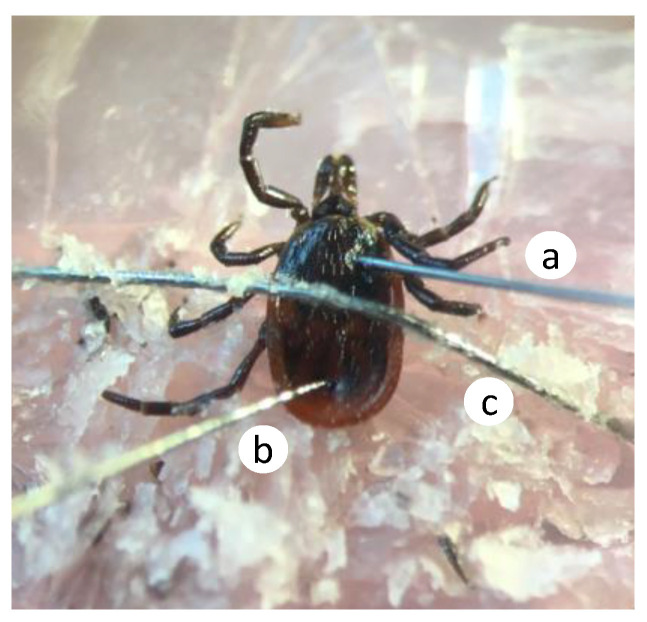
Dorsal view of the tick preparation for electrophysiological recordings. Recording electrode (**a**) was inserted into the scutum, and ground electrode (**b**) into the alloscutum, while the minuten pin (**c**) holds the tick in place.

**Figure 2 insects-11-00502-f002:**
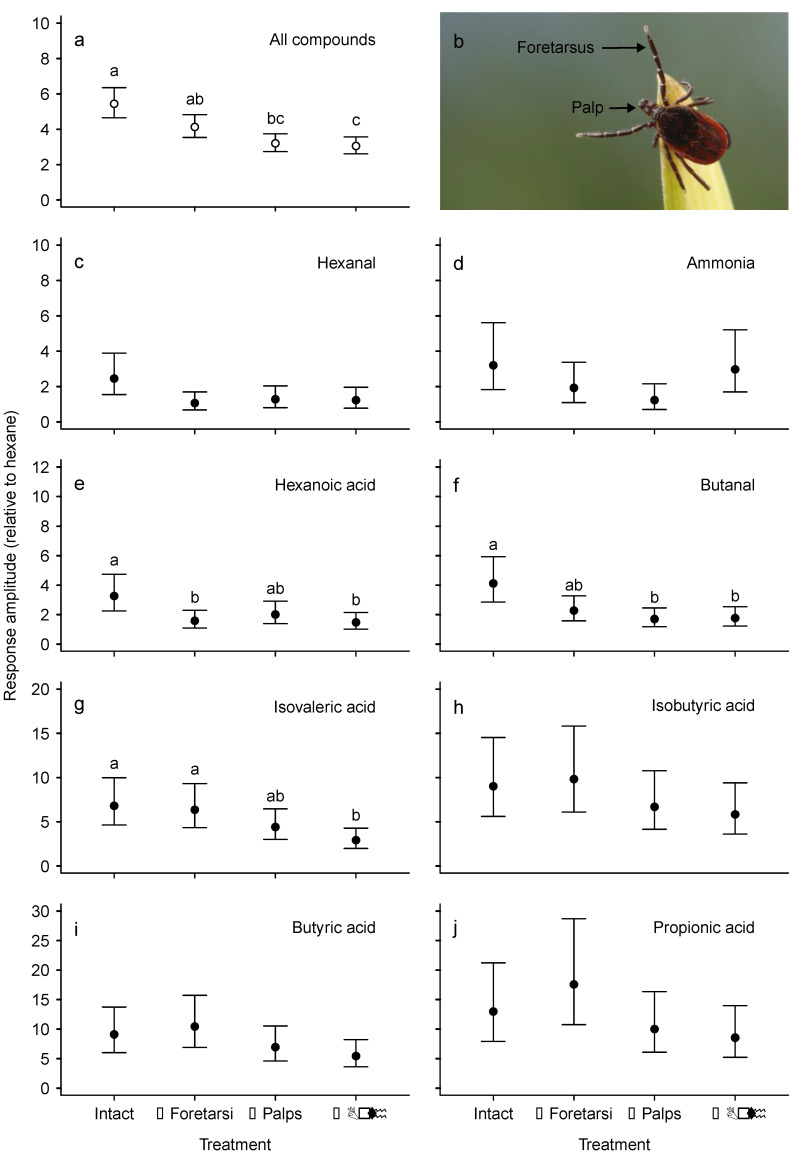
Electroscutumography (ESG) responses of intact ticks and ticks with appendages (foretarsi, palps, or both) surgically removed (*N* = 21 for each treatment) that were presented with a series of attractants (100 μg/μL) that elicited responses significantly different from those to hexane (see Table 2). (**a**) Effects of surgery on tick responses across all compounds. (**b**) A questing tick, illustrating the foretarsi and palps (photo: Sean McCann). (**c**–**j**) Effects of surgery on tick responses to individual compounds. Response amplitude was calculated relative to hexane (control) by dividing the responses of each tick by its mean response to three hexane puffs (at the beginning, middle, and end of each run). Back-transformed marginal means (points) and 95% confidence intervals (error bars) estimated from individual ANOVAs with log (relative response) as the response variable and surgical treatment as a fixed predictor variable are shown for each compound. Different letters indicate significant differences using Tukey’s method for multiple comparisons.

**Figure 3 insects-11-00502-f003:**
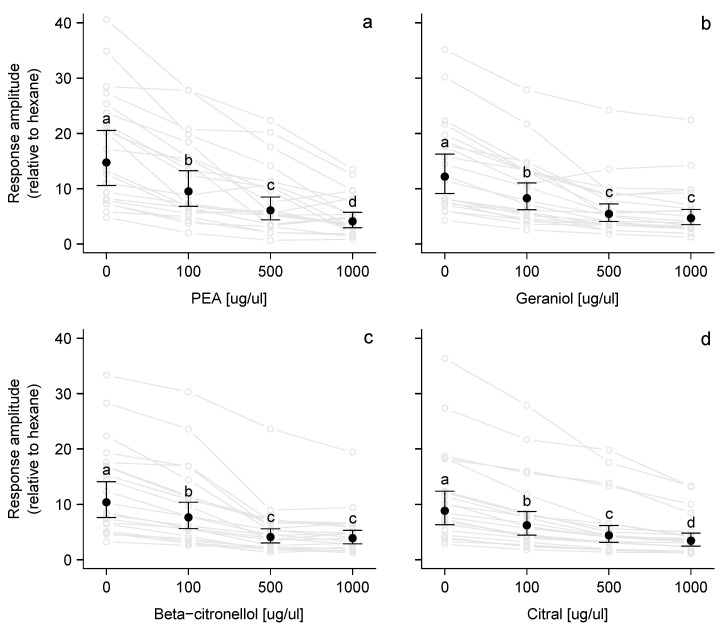
Mean electroscutumography (ESG) responses of ticks (*N* = 21) presented with butyric acid (BTA, 100 μg/μL) alone and in combination with increasing doses (100–1000 μg/μL) of a VOC: (**a**) phenethyl alcohol (PEA), (**b**) geraniol, (**c**) β-citronellol, and (**d**) citral. Response amplitude was calculated relative to hexane by dividing the responses of each tick by its mean response to hexane (control) puffs. Plots show the raw data in grey (lines connect responses from individual ticks) and in black, the back-transformed marginal means and 95% confidence intervals estimated from models that included log(relative response) as the response variable and tick number as a random blocking factor (see methods for details). Different letters indicate significant differences using Tukey’s method for multiple comparisons.

**Figure 4 insects-11-00502-f004:**
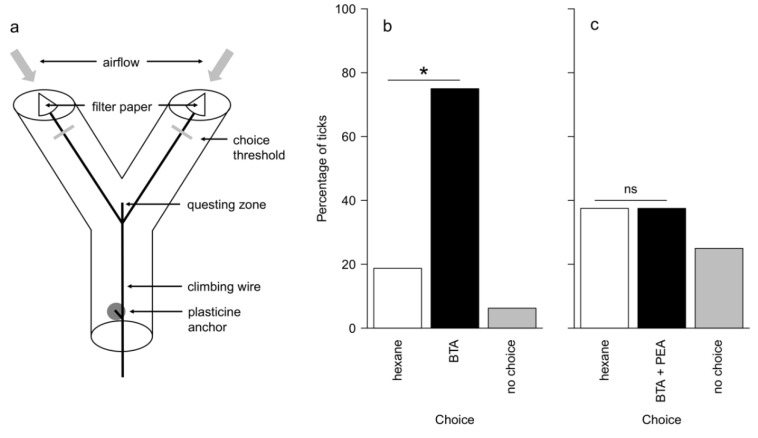
(**a**) Y-tube behavioural assays set-up. The filter papers were loaded with hexane (control) or the stimulus; Results of vertical Y-tube bioassays testing the attraction of unfed adult female *I. scapularis* to (**b**) butyric acid (BTA, 10 μg/μL) and to (**c**) the binary mixture of BTA and phenethyl alcohol (BTA 10 μg/μL + PEA 10 μg/μL) (*N* = 16). Hexane was used as control. No-choice bar represents the number of non-responders. The percentage of ticks choosing the treatment arm was compared to 50% (null expectation) using a binomial test; * = *p* < 0.05, ns = not significant (*p* > 0.05).

**Table 1 insects-11-00502-t001:** Stimuli presented during electroscutumography (ESG) experiments. Butyric acid (BTA) stimulus was first applied to the stimulus cartridge, followed by the application of the VOC at a given concentration.

Treatment	Concentration BTA	Concentration VOC ^†^
control	0 *	0 *
0	100 μg/μL	0 *
100	100 μg/μL	100 μg/μL
500	100 μg/μL	500 μg/μL
1000	100 μg/μL	1000 μg/μL

* hexane; ^†^ PEA, geraniol, β-citronellol, or citral.

**Table 2 insects-11-00502-t002:** Electroscutumography (ESG) responses of intact ticks (*N* = 21) to a series of attractant VOCs. Responses are displayed as marginal means and 95% confidence intervals estimated from a model that included tick number as a random factor to account for repeated measurements of each individual. The responses were log-transformed before the model and post-hoc tests, and here we display back-transformed estimates on the original scaled measurement (in mV).

VOC	Number of Carbons	^1^ Functional Groups	mV	95% CI	Tukey’s
^2^*n*-Hexane	6	SH	2.0	(1.2, 3.3)	a
Pentanal	5	SH; A	1.8	(1.1, 2.9)	a
Octanal	8	SH; A	1.8	(1.1, 3.0)	a
Lactic acid	3	SH; CA; H	2.1	(1.3, 3.4)	a
Heptanal	7	SH; A	2.6	(1.6, 4.3)	a
Hexanal	6	SH; A	5.0	(3.1, 8.1)	b
NH_3_	-	-	6.5	(4.0, 10.6)	bc
Hexanoic acid	6	SH; CA	6.6	(4.1, 10.8)	bc
Butanal	4	SH; A	8.4	(5.1, 13.6)	c
Isovaleric acid	5	SH; CA	13.8	(8.5, 22.6)	d
Isobutyric acid	4	SH; CA	18.3	(11.2, 29.9)	de
Butyric acid	4	SH; CA	18.5	(11.3, 30.2)	de
Propionic acid	3	SH; CA	26.4	(16.1, 43.0)	e

^1^ A = aldehyde (-COH); H = hydroxyl (-OH); CA = carboxylic acid (-COOH); SH = saturated hydrocarbon. ^2^ Three hexane (control) puffs were presented at the beginning, middle, and end of each series of recordings, and the results were averaged for each tick.

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
