# Peer review of "Chemosensory and Behavioural Responses of Ixodes scapularis to Natural Products: Role of Chemosensory Organs in Volatile Detection"

_insects, 2020, doi:10.3390/insects11080502_

Round 1

Reviewer 1 Report

This paper presents some interesting findings,  adding to our current understand of the detection of volatiles by ticks and certainly merits publication.

However, I believe the introduction needs to be cut down…the discussion of the existing literature could be reduced, just presenting a resume of what we know and outlining the goals of the study.

Similarly, the discussion provides a lot of information that should have been presented earlier, as it actually explains the reasoning behind the experiments that were carried out, so possibly include a somewhat briefer description the Methods. As an example, in the first paragraph the authors provide the background for where they chose to insert the electrode. There is also a repeat of results in the discussion that is not really necessary

Is there any information on the age of the ticks, or of their reproductive state (mated or not) as these factors could affect responses?

It is unfortunate that when testing responses following the excision of the palps/fore-tarsi there was not a control where another body part not involved in the detection of infochemicals (such as part of a back leg) was removed: this would allow one to eliminate any direct effect of just having a body part amputated.

Check text and reference for minor errors. For example, in ln 351 I believe it should be concentration rather than concertation, and in reference 10 the scientific name is not in italics

Author Response

Reviewer 1

This paper presents some interesting findings, adding to our current understand of the detection of volatiles by ticks and certainly merits publication.

However, I believe the introduction needs to be cut down…the discussion of the existing literature could be reduced, just presenting a resume of what we know and outlining the goals of the study.

We thank Reviewer 1 for the comments. We believe the introduction introduces well the status of the research on tick chemosensory highlighting the gaps that need to be filled. In our opinion this novel approach requires a proper and complete introduction.

Similarly, the discussion provides a lot of information that should have been presented earlier, as it actually explains the reasoning behind the experiments that were carried out, so possibly include a somewhat briefer description the Methods. As an example, in the first paragraph the authors provide the background for where they chose to insert the electrode. There is also a repeat of results in the discussion that is not really necessary.

We removed the sentence repeating the background on electrode insertion.

Is there any information on the age of the ticks, or of their reproductive state (mated or not) as these factors could affect responses?

Adult ticks were about a month old, unmated and not engorged. We have included the term “adult”.

It is unfortunate that when testing responses following the excision of the palps/fore-tarsi there was not a control where another body part not involved in the detection of infochemicals (such as part of a back leg) was removed: this would allow one to eliminate any direct effect of just having a body part amputated.

We have not removed part of the 4th legs as control because from our previous experiments (Faraone et al. 2019, Behavioral responses of Ixodes scapularis tick to natural products: development of novel repellents. Exp. Appl. Acarol., 2019, 79, 195-207), the surgical removal of the 4th legs did not affect tick behavior in response to volatiles.

Check text and reference for minor errors. For example, in ln 351 I believe it should be concentration rather than concertation, and in reference 10 the scientific name is not in italics

We thank the Reviewer for catching this! We have corrected the spelling mistake.

Reviewer 2 Report

The paper presented entitled "Chemosensory and behavioral responses of Ixodes scapularis to natural products: role of chemosensory organs in volatile detection" by Nicoletta Faraone et al is interesting. Provides valuable new information on the behavior of Ixodes scapularis ticks. Introduction provide sufficient background and include relevant references. The research design is appropriate, however, the description of the methods requires clarification (see detailed comments). I suggest minor revision.

Detailed comments:

L87 In my pdf version of the manuscript I see “Naïve, unfed, host-seeking”. My guess the authors meant "Native"

L 111 instead of using the word “idiosoma” I suggest using the word “alloscutum”. (In fact, scutum is part of idiosoma).

L 102 are the units correct? mg/mL or mg/ml? In addition to L 102 and elsewhere, please use consistent unit notation. See L104, 107, 134, 135, 139. Also, if you are giving the results of statistical tests, use one way of writing the result with a space after the "p" or without, see L 243, 264, 283 and everywhere else. Please follow consistency

L 303 .... to several odor stimuli.  The syn ganglion represents ... No extra space needed between sentences

L 330. This is the first time you mention Amblyomma variegatum in the text. Please, write full name of the genus and species instead of the abbreviation.

Figure 2. I suggest enlarging the graphs similar to Figure 3. You can do this by changing the orientation or placing only 2 graphs in one row instead of 3.

My greatest misunderstanding. Please explain the number of ticks used in each experiment in the description of the methodology. Now it is incomprehensible to the reader. You give 15 one time, another 16 or 21.

My next methodological question is whether the experiment uses ticks of the same physiological age?

Author Response

Reviewer 2

The paper presented entitled "Chemosensory and behavioral responses of Ixodes scapularis to natural products: role of chemosensory organs in volatile detection" by Nicoletta Faraone et al is interesting. Provides valuable new information on the behavior of Ixodes scapularis ticks. Introduction provide sufficient background and include relevant references. The research design is appropriate, however, the description of the methods requires clarification (see detailed comments). I suggest minor revision.

We thank Reviewer 2 for the comments and we are pleased that our work is found to be interesting.

Detailed comments:

L87 In my pdf version of the manuscript I see “Naïve, unfed, host-seeking”. My guess the authors meant "Native"

With the term “naïve” we mean that ticks have never been exposed to any tested volatile before so they were “inexperienced”.

L 111 instead of using the word “idiosoma” I suggest using the word “alloscutum”. (In fact, scutum is part of idiosoma).

We have changed “idiosoma” with “alloscutum”.

L 102 are the units correct? mg/mL or mg/ml? In addition to L 102 and elsewhere, please use consistent unit notation. See L104, 107, 134, 135, 139. Also, if you are giving the results of statistical tests, use one way of writing the result with a space after the "p" or without, see L 243, 264, 283 and everywhere else. Please follow consistency

Concentrations are expressed in ug/ul and we have corrected “ml” and we have included the space after “p”

L 303 .... to several odor stimuli.  The syn ganglion represents ... No extra space needed between sentences

We have corrected it.

L 330. This is the first time you mention Amblyomma variegatum in the text. Please, write full name of the genus and species instead of the abbreviation.

We thank Reviewer 2 for catching this. We have reported the full name.

Figure 2. I suggest enlarging the graphs similar to Figure 3. You can do this by changing the orientation or placing only 2 graphs in one row instead of 3.

We have provided a new version of the figure with 2 plots per row.

My greatest misunderstanding. Please explain the number of ticks used in each experiment in the description of the methodology. Now it is incomprehensible to the reader. You give 15 one time, another 16 or 21.

Thank you for catching this error. Where N=15 was indicated in the results, it should have read N=21. We have corrected the sample sizes where this error occurred. We did use different numbers of ticks in the 3 different  experiments based on tick availability (N=16 for each Y-tube assay and N=21 for each electrophysiological series). We now state the corrected sample sizes for each experiment in both the methods and results, and we have added a sentence (lines 88-89) to indicate that each tick was used in only one experiment, to help clarify total sample sizes (a total of 74 ticks were used in the study).  

My next methodological question is whether the experiment uses ticks of the same physiological age?

This is correct. Used ticks were from the same age and status. Adult ticks were about a month old, unmated and not engorged. We have included the term “adult”.

Reviewer 3 Report

The manuscript is well written, concise and descriptive. It assessed the behavioural and chemosensory responses of Ixodes scapularis female ticks using the known Y-tube bioassay to monitor behavioural responses to known attractants and repellents. This has been reported for ticks in previous studies along with neurological involvement (synganglion). In the current study, neurological responses were measured in the synganglion with the use of a probe inserted through the scutum in an attempt to establishing an underlying neurophysiological mechanism to tick olfaction in the absence of olfactory organs (palps and Haller's organ). Increasing concentrations of repellent along with, and separate to, constant concentrations of attractant stimulant were evaluated confirming an inhibitory action to the ability of the tick olfactory system to detect volatile attractants at higher repellent concentrations. It was confirmed that both the palps and Haller's organs are involved in the detection of volatiles but suggested palps more so, which is contradictory to previous studies. This might also be of interest in further research to establish the possible difference in receptors of generalist feeders vs. host specific tick species.

Although the authors did make use of humidified air before delivering the stimulus, it is along with heat sensing, not considered as "classical" chemo-sensing in ticks. The flip side of the coin is that ticks' questing behaviour is stimulated when they are dehydrated. Nonetheless, one can only wonder how the addition of a heat source might have influenced their behaviour as well.

Overall, an interesting study nonetheless.

A few notes:
I suggest moving figure 1 to after line 121 where the first reference to the figure was made.
Lines 133-136 - just for clarification, was the tick mounted glass slide placed inside the 30cm tube or how was the stimulus applied to the tick? If the setup was adapted from Romanshchenko et al., is it possible to include a diagram of your adaption? Maybe combine it with figure 1?
Line 351 - ...concentration...
Line 365-367 - As the author noted in their previous studies; they needed the tick to perceive the presence of the observer to elicit a locomotory response, the integrated biology of the tick as a whole has to be considered here. They have eyes, motion and heat sensors as well that influence their questing behaviour and might not be solely due to concentration range of attractants. These ranges might be minimal and possibly skewed beyond physiological relevant concentration ranges for the purpose of getting the tick to move. If the tick is allowed to see /'feel' the host approaching as their olfactory systems confirms what the sensory systems perceive (an approaching host), then concentration ranges might differ as well (with implications for repellent concentrations as well).

Author Response

Reviewer 3

The manuscript is well written, concise and descriptive. It assessed the behavioural and chemosensory responses of Ixodes scapularis female ticks using the known Y-tube bioassay to monitor behavioural responses to known attractants and repellents. This has been reported for ticks in previous studies along with neurological involvement (synganglion). In the current study, neurological responses were measured in the synganglion with the use of a probe inserted through the scutum in an attempt to establishing an underlying neurophysiological mechanism to tick olfaction in the absence of olfactory organs (palps and Haller's organ). Increasing concentrations of repellent along with, and separate to, constant concentrations of attractant stimulant were evaluated confirming an inhibitory action to the ability of the tick olfactory system to detect volatile attractants at higher repellent concentrations. It was confirmed that both the palps and Haller's organs are involved in the detection of volatiles but suggested palps more so, which is contradictory to previous studies. This might also be of interest in further research to establish the possible difference in receptors of generalist feeders vs. host specific tick species.

Although the authors did make use of humidified air before delivering the stimulus, it is along with heat sensing, not considered as "classical" chemo-sensing in ticks. The flip side of the coin is that ticks' questing behaviour is stimulated when they are dehydrated. Nonetheless, one can only wonder how the addition of a heat source might have influenced their behaviour as well.

We thank Reviewer 3 for the valuable comments. This is a very interesting point and we are currently evaluating through behavioral and electrophysiological studies the impact of temperature of tick response to volatiles as well.

Overall, an interesting study nonetheless.

A few notes:
I suggest moving figure 1 to after line 121 where the first reference to the figure was made.

We have moved Figure 1 as suggested.

Lines 133-136 - just for clarification, was the tick mounted glass slide placed inside the 30cm tube or how was the stimulus applied to the tick? If the setup was adapted from Romanshchenko et al., is it possible to include a diagram of your adaption? Maybe combine it with figure 1?

The mounted tick was located in front of the outlet of a 30 cm long glass tube. This tube has a single 2 mm diameter hole 10 cm from the outlet from where a glass pipette containing the stimulus cartridge was inserted to deliver the stimulus. This setup is commonly sued in EAG with insects. We hope this description clarifies any doubts.

Line 351 - ...concentration...

Thanks for catching this. We have corrected the misspelling.

Line 365-367 - As the author noted in their previous studies; they needed the tick to perceive the presence of the observer to elicit a locomotory response, the integrated biology of the tick as a whole has to be considered here. They have eyes, motion and heat sensors as well that influence their questing behaviour and might not be solely due to concentration range of attractants. These ranges might be minimal and possibly skewed beyond physiological relevant concentration ranges for the purpose of getting the tick to move. If the tick is allowed to see /'feel' the host approaching as their olfactory systems confirms what the sensory systems perceive (an approaching host), then concentration ranges might differ as well (with implications for repellent concentrations as well).

We thank the Reviewer for these observations. Indeed, there are other important parameters that might influence tick response. We are currently designing other experiments in order to evaluate physiological variables that are contributing to the overall response to chemical cues.